# SARS-COV-2 Omicron variants conformationally escape a rare quaternary antibody binding mode

Jule Goike[1,2,14], Ching-Lin Hsieh [2,14], Andrew P. Horton [1,2,3,14], Elizabeth C. Gardner[1,2,14], Ling Zhou[2], Foteini Bartzoka[1,2], Nianshuang Wang[2], Kamyab Javanmardi[1,2], Andrew Herbert[4], Shawn Abbassi[4], Xuping Xie [5], Hongjie Xia [5], Pei-Yong Shi [5], Rebecca Renberg [6], Thomas H. Segall-Shapiro[3,7], Cynthia I. Terrace[7], Wesley Wu[3], Raghav Shroff[1,2,7], Michelle Byrom[1], Andrew D. Ellington [1,2,8], Edward M. Marcotte [1,2], James M. Musser[3], Suresh V. Kuchipudi [9], Vivek Kapur[10], George Georgiou [1,2,8,11,12], Scott C. Weaver [13], John M. Dye[4], Daniel R. Boutz [1,2,3,7✉], Jason S. McLellan [2✉] & Jimmy D. Gollihar [1,2,3,7✉]

The ongoing evolution of SARS-CoV-2 into more easily transmissible and infectious variants has provided unprecedented insight into mutations enabling immune escape. Understanding how these mutations affect the dynamics of antibody-antigen interactions is crucial to the development of broadly protective antibodies and vaccines. Here we report the characterization of a potent neutralizing antibody (N3-1) identified from a COVID-19 patient during the first disease wave. Cryogenic electron microscopy revealed a quaternary binding mode that enables direct interactions with all three receptor-binding domains of the spike protein trimer, resulting in extraordinary avidity and potent neutralization of all major variants of concern until the emergence of Omicron. Structure-based rational design of N3-1 mutants improved binding to all Omicron variants but only partially restored neutralization of the conformationally distinct Omicron BA.1. This study provides new insights into immune evasion through changes in spike protein dynamics and highlights considerations for future conformationally biased multivalent vaccine designs.

[1] Center for Systems and Synthetic Biology, Department of Molecular Biosciences, The University of Texas at Austin, Austin, TX, USA. [2] Department of Molecular Biosciences, The University of Texas at Austin, Austin, TX, USA. [3] Antibody Discovery and Accelerated Protein Therapeutics, Department of Pathology and Genomic Medicine, Houston Methodist Research Institute, Houston Methodist Hospital, Houston, TX, USA. [4] U.S. Army Medical Research Institute of Infectious Diseases, Frederick, MD, USA. [5] Department of Biochemistry and Molecular Biology, University of Texas Medical Branch, Galveston, TX, USA. [6] DEVCOM Army Research Laboratory, Biotechnology Branch, Adelphi, MD, USA. [7] DEVCOM Army Research Laboratory-South, Austin, TX, USA. [8] Department of Chemistry, The University of Texas at Austin, Austin, TX, USA. [9] Department of Veterinary and Biomedical Science and Animal Diagnostic Laboratory, The Pennsylvania State University, University Park, PA, USA. [10] Department of Animal Science and the Huck Institutes of the Life Sciences, The Pennsylvania State University, University Park, PA, USA. [11] Department of Chemical Engineering, The University of Texas at Austin, Austin, TX, USA. [12] Department of Oncology, Dell Medical School, The University of Texas at Austin, Austin, TX, USA. [13] University of Texas Medical Branch, World Reference Center for Emerging Viruses and Arboviruses, Galveston, TX, USA. [14] These authors contributed equally: Jule Goike, Ching-Lin Hsieh, Andrew P. Horton, Elizabeth C. Gardner. ✉email: drboutz@houstonmethodist.org; jmclellan@austin.utexas.edu; jgollihar2@houstonmethodist.org

The rapid global dissemination of the severe acute respiratory syndrome coronavirus 2 (SARS-CoV-2)[1], the cause of coronavirus disease 2019 (COVID-19)[2], highlights our extreme vulnerability to novel pathogens. The speed of SARS-CoV-2 transmission and the absence of widespread adaptive immunity created a pandemic that overwhelmed the international medical community. This situation has been exacerbated by the scarcity of treatment options throughout variant-dominated waves of the pandemic[3].

SARS-COV-2 Alpha, Beta, Gamma, and Delta each accumulated a distinct set of spike protein mutations and were considered variants of concern (VOCs) due to their increased transmissibility[4–8] before the emergence and rapid dissemination of the Omicron lineages[9]. Omicron spike proteins have accumulated an unprecedented number of mutations in the N-terminal domain (NTD), receptor binding domain (RBD), fusogenic stalk (S2), and furin cleavage site. While previous VOCs primarily showed escape of NTD-directed antibodies[10], Omicron's extensive RBD mutations enabled it to evade pre-existing RBD binders, including complete escape from therapeutic antibodies targeting the receptor binding motif, such as REGN10933, REGN10987, LY-CoV555 and LY-CoV016 which had received FDA emergency use authorization[11].

Interestingly, Omicron's immune evasion cannot solely be attributed to amino acid substitutions disrupting antibody epitopes. Instead, altered RBD conformational dynamics contribute to the antigenic drift observed by evading conformational binders[12,13]. The spike protein trimer is a metastable molecule in which the three RBD domains can adopt 'up' or 'down' conformations. While WHU1 spike is most often observed in the one-RBD-up state, the RBDs are conformationally flexible and able to adopt multiple configurations[14]. However, the Omicron lineage displays a shift towards strictly stabilizing a two down, one up (BA.1) or three down (BA.2 and BA.5) state[12,15,16].

We developed a multi-pronged antibody discovery and informatics strategy that combines proteomic analysis of donor sera with the selection of combinatorial paired heavy and light chain (VH-VL) libraries from donor B-cell receptor repertoires. This integrative approach allowed us to probe both the secreted circulating antibody repertoire and the nascent cellular repertoire of primary immune responses, and we identify potent neutralizing antibodies from two convalescent donors infected during the first wave of COVID-19. We structurally characterized a highly potent antibody (N3-1) that binds a quaternary epitope of the trimeric spike protein via a RBD binding modality. N3-1 maintained robust neutralization against all VOCs prior to the emergence of Omicron. We show that the altered RBD up-down equilibria of Omicron is primarily responsible for its escape from N3-1, and we discuss implications for rational vaccine design.

## Results

**IgSeq analysis of the serological repertoire reveals candidate heavy chains**. IgSeq proteomics provides a snapshot of the serum antibody repertoire by using mass spectrometry to identify heavy chain complementarity-determining region 3 (CDR3) peptides of antigen-enriched antibodies (Fig. 1a)[17]. To identify relevant SARS-CoV-2 antibodies present in the serological repertoire, we first isolated antibodies from the serum of two donors by antigen enrichment chromatography using immobilized SARS-CoV-2 RBD and ectodomain (ECD) fragments. We then employed IgSeq proteomics to identify abundant IgG clonotypes, sets of clonally related VH sequences that share similar or identical CDR3s. A total of 15 and 21 unique clonotypes were identified with high confidence from donors 1 and 2, respectively, and we selected a representative full VH sequence for each (Supplementary Fig. 1, 2).

**Parallel selections *via* yeast surface display yield neutralizing antibodies**. Much of the early-stage primary antibody response is inaccessible to proteomic discovery, as detection is limited to secreted antibody proteins with sufficient abundance in serum[18]. To broaden our search for antigen-specific VH and neutralizing antibodies, we developed a humanized yeast surface display (YSD) selection platform to directly mine the cellular repertoire for high affinity B-cell receptor sequences. We constructed a large Fab library from combinatorially assembled donor VH-VL pairs, transformed it into a humanized yeast strain, and selected for Fab binding and expression over multiple rounds of fluorescent cell sorting (Fig. 1d, Supplementary Figs. 3-5). We performed three rounds of enrichment and identified and quantified productive VH-VL pairs using MinION nanopore sequencing of the Fab amplicons (Supplementary Figs. 6-7). Likewise, we performed YSD selections on a large Fab library constructed from IgSeq-defined VH and donor-derived VL (Fig. 1b). Using YSD, a total of 9 and 17 neutralizing mAbs were identified by full combinatorial screening of donor-derived VH-VL libraries and IgSeq-VH donor-VL libraries, respectively (Supplementary Table 1).

**Public light chains recover productive VH-VL pairs**. While IgSeq proteomics provides valuable information for antibody heavy chain discovery, it does not identify light chain partners, and requires additional laborious techniques. Further, the combinatorial pairing of VH-VL in YSD may not yield optimal antibody candidates due to limitations with library transformation. To expedite light chain discovery, we analyzed a published dataset of natively paired memory B-cell sequences[19] from three healthy donors and bioinformatically derived a panel of nine light chains that show high frequencies of productive pairings with a diverse set of VH sequences (Figs. 1c, 2a).

We hypothesized that this abundance could indicate an enhanced ability to form productive VH-VL pairs independent of VH sequence and antigen specificity. To test this, we constructed germline versions of these light chains with their most frequently observed J gene[20–23] (Supplementary Table 2) to obtain full-length VL for each public light chain (PLC). 22 SARS-CoV-2-specific heavy chains identified through IgSeq and/or YSD were then expressed with each PLC as a full-length IgG1. $EC_{50}$ determination via ELISA using recombinant spike protein demonstrated that all included VHs paired productively with one or more of the nine PLCs, resulting in several highly potent and specific antibodies (Fig. 2b).

We recovered 15 neutralizing VH-PLC pairs from 8 unique VHs (Fig. 2b-c, Supplementary Table 1). Interestingly, NTD-directed VHs showed strong preference for lambda PLCs, most commonly PLC-8 (IGLV1-51). We also observed that four VHs produced neutralizing antibodies with more than one PLC. The VH "N3", for example, was a highly potent RBD-directed binder, showing the greatest neutralization potency with PLC-1 ($IC_{50} = 0.25$ nM) and forming additional neutralizing pairs with PLC-3 ($IC_{50} = 13.1$ nM) and PLC-7 ($IC_{50} = 27.7$ nM). Structural analysis of N3-1 in complex with the spike protein revealed that PLC-1 both stabilizes N3 and actively takes part in binding to a quaternary epitope (Fig. 3).

**N3-1 exhibits a quaternary binding mode**. Structurally defined RBD-targeted antibodies and nanobodies complexed with SARS-CoV-2 S-protein may be generally grouped into four classes, as described by Barnes et al.[24]. The class I and II antibodies mainly derive from germline gene IGHV3-53 and recognizes the angiotensin converting enzyme 2 (ACE2) -binding site, directly blocking host receptor engagement (e.g. antibodies CC12[25], C105[26] and VHH E[27]). The class III and IV antibodies, with more

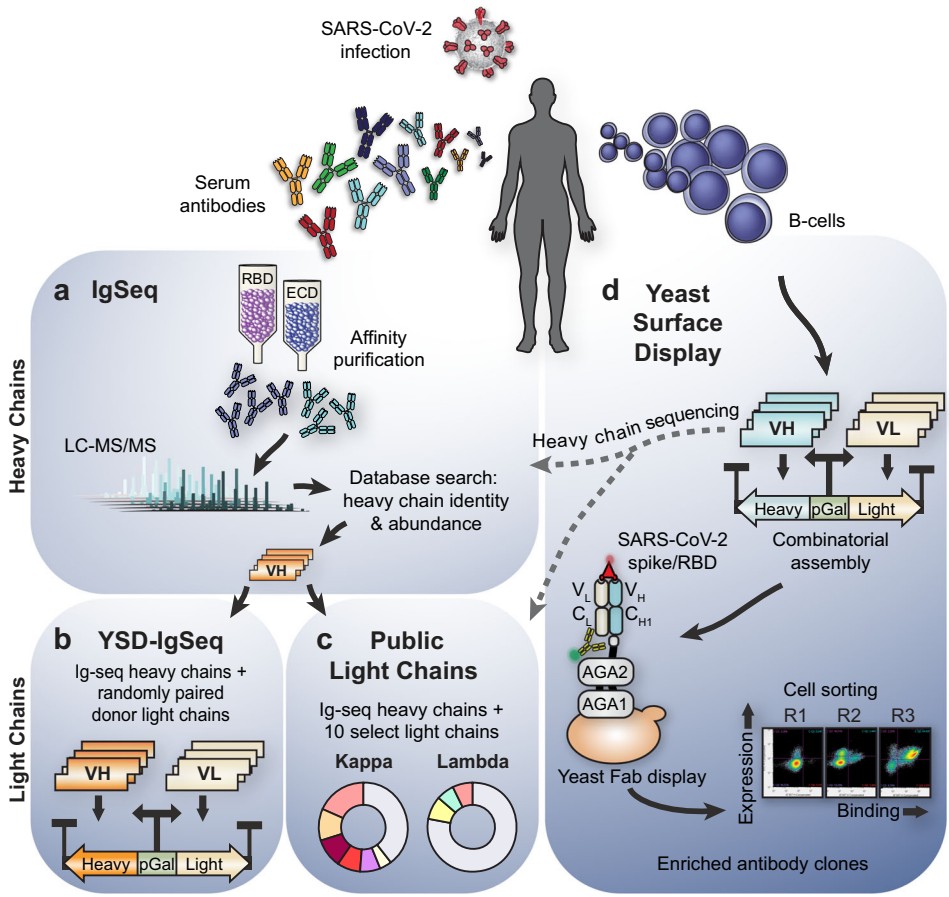

**Fig. 1 Overview of complementary strategies for discovering convalescent patient-derived anti-SARS-CoV-2 neutralizing antibodies.** Serum antibodies and B-cells are isolated from patient blood postinfection. **a** IgG is antigen-enriched, digested, and analyzed with tandem mass spectrometry. Searching against a database of donor B cell receptor sequences yields antigen-specific heavy chain candidates. **b** To find productive VH-VL pairs, the proteomically identified VH are randomly combined with donor VL, generating a large library of Fab amplicons for selection in yeast. **c** Additional productive VH-VL are recovered by matching each proteomic VH candidate against a fixed set of nine public light chains significantly overrepresented in prior repertoire studies. All combinations are screened for binding. **d** Yeast surface display provides an independent method to recover VH or VH-VL candidates. VH and VL sequences are amplified from donor B-cells, and pairs are combinatorially assembled into large Fab libraries and selected using fluorescent cell sorting.

diverse IGHV gene usage, recognize a relatively conserved region on the RBD that is mostly buried and contacts a neighboring RBD of the S-protein trimer in the 'down' conformation (e.g. C135[24], S309[28], and VHH V[27]). From negative stain EM analysis of the IgG N3-1-spike complex we observed two Fabs, likely derived from the same IgG molecule, bound to multiple RBDs on a single trimeric spike (Supplementary Fig. 8). We also performed SEC-MALS for the N3-1 IgG in complex with SARS-CoV-2 spike, and the measured molecular weights show 1:1 binding of IgG to spike and an excess of free IgG, supporting our proposed model (Supplementary Fig. 9).

To investigate this unique binding mode, we determined a cryo-EM structure of N3-1 Fab bound to SARS-CoV-2 S-ECD to a global resolution of 2.8 Å (Fig. 3a). We observed that two N3-1 Fabs bind to a single trimeric spike with one Fab binding to RBD in the 'up' conformation and the other Fab simultaneously engaging two RBDs: one in the 'up' conformation and one in the 'down' conformation. We performed focused refinement on the Fab bound to the two RBDs (Fig. 3b), which substantially improved the interpretability of the map in this region. The well-resolved Fab-RBD binding interface revealed two completely distinct epitopes on RBD-up and RBD-down (Fig. 3c). We also performed focused refinement on the Fab bound to the single RBD in the 'up' conformation (Fig. 3d). Superimposition of two focused maps reveals that both Fabs make the same set of

contacts with respect to RBD-up, with an RMSD of 0.7 Å over 380 Cα atoms (Supplementary Fig. 8c). Furthermore, superimposing the N3-1 Fab bound to RBD-up structure with an apo spike structure that has two RBDs in the down conformation shows clashes, indicating N3-1 only binds spike with two RBDs in the up conformation (Supplementary Fig. 8e).

For the RBD-up interaction, contacts are made by the Fab H-CDR1, H-CDR3 and L-CDR2, which together bury 824 Å² surface area (Fig. 3c). For the RBD-down interaction, contacts are made by the Fab H-CDR2, H-CDR3 and L-CDR3, which buries 709 Å² surface area (Fig. 3e). Notably, the relatively long H-CDR3 loop (ARGTIYFDRSGYRRVDPFHI, 20 a.a., residues 93-99,100a-100j,101-102 in the Kabat antibody numbering scheme) engages both RBDs via hydrophobic and polar interactions (Fig. 3c). H-CDR3 residues Tyr98, Phe99 and Arg100a pack against a hydrophobic pocket formed by Tyr369, Phe377, Lys378 and Pro384 on RBD-up (Fig. 3c), which are highly conserved between SARS-CoV and SARS-CoV-2. This pocket is barely exposed when RBD is in the 'down' conformation and is part of the shared epitopes targeted by SARS-CoV and SARS-CoV-2 cross-reactive antibodies CR3022 and COVA1-16[29]. Lys378, from the upper ridge of the pocket, forms a cation-π interaction with Tyr98 of H-CDR3, and its amine group is expected to form a salt bridge with Asp100h of H-CDR3 (Fig. 3c). In addition, main chain atoms of Cys379 and Tyr369 form hydrogen bonds with Phe99 and

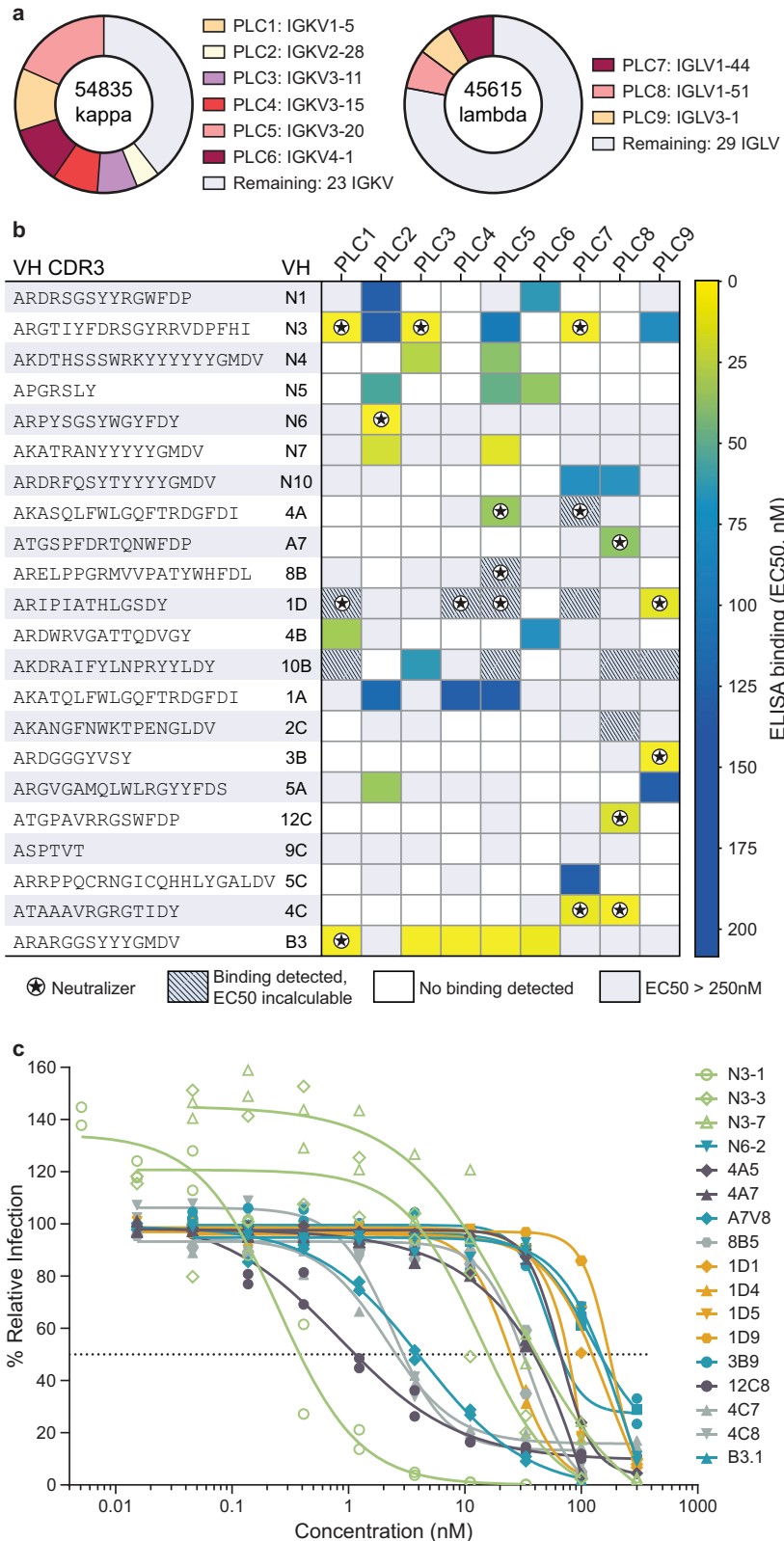

**Fig. 2 Public light chain screening and neutralization. a** Nine PLCs emerge from analysis of 100,450 previously published paired VH-VL sequences. **b** Screening with VHs (H-CDR3 depicted) identified by IgSeq and YSD against the panel of nine PLCs to determine productive VH-VL pairings. IgG mAbs ELISA $EC_{50}$s revealed that partnering VHs with PLCs can create low nM affinity binders. Grey boxes indicate that binding was detected but at an affinity too low to determine an $EC_{50}$ in the concentration range tested. White boxes, no binding was detected. **c** 17 VH-PLC pairs from (**b**) neutralize SARS-CoV-2 WA1 virus, fit to two biological replicates.

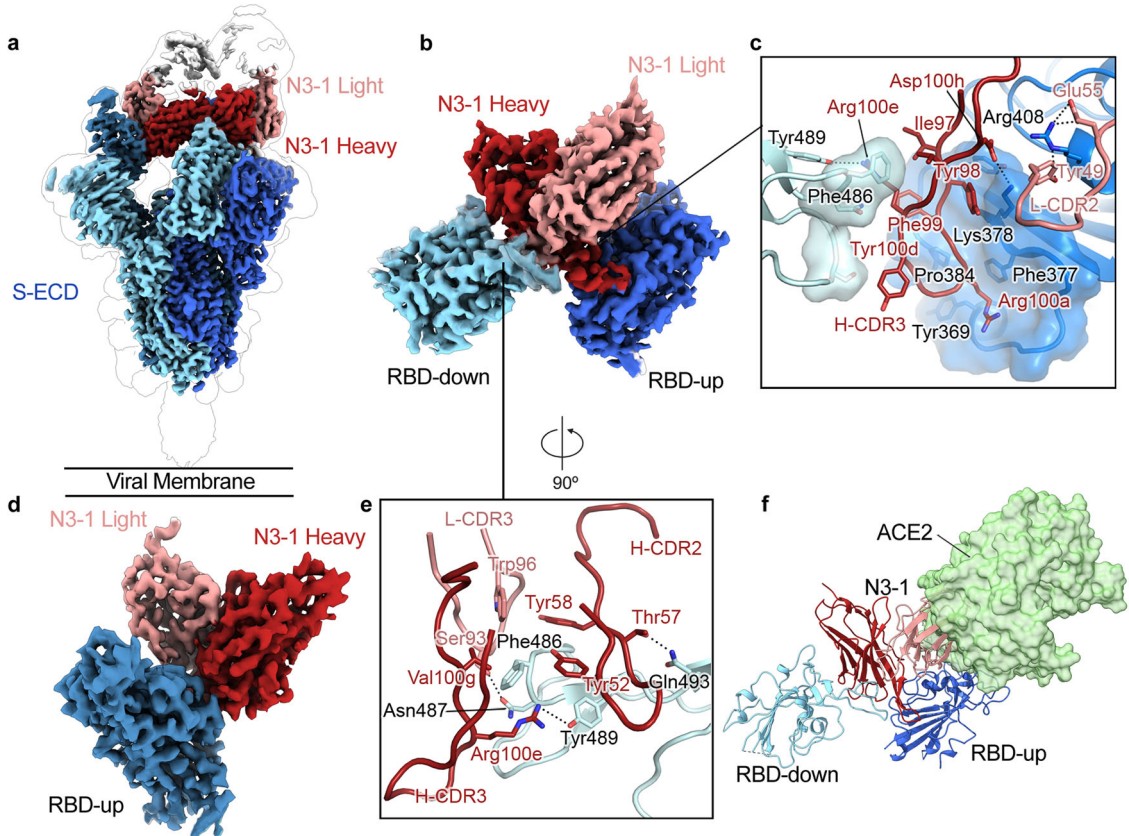

**Fig. 3 RBD-directed mAb N3-1 exhibits a unique binding mode by recognizing two distinct epitopes. a** Composite cryo-EM map of N3-1-bound SARS-CoV-2 Wuhan-Hu-1 spike. Each protomer is depicted in steel blue, royal blue and sky blue. The heavy chain of N3-1 is colored firebrick, and the light chain is colored coral. A lower resolution contour is shown to help visualize the Fabs and the RBDs. **b** Focused refinement map of N3-1 bound to RBDs in the up and down conformations. **c** One face of H-CDR3 contacts a conserved hydrophobic pocket (transparent royal blue surface) on RBD-up. The other face of H-CDR3 contacts the ACE2-binding site on RBD-down. H-CDR1 contacts the epitopes on RBD-up, but it is omitted for clarity. **d** Focused refinement map of N3-1 bound to RBD in the up conformation. **e** The epitope on RBD-down is centered on Phe486, which fits into a hydrophobic surface formed by Trp96, Tyr58, Tyr52, Arg100e and Val100g (clockwise). **f** Superimposed crystal structure of RBD-ACE2 complex (PDB ID: 6M0J) with N3-1 bound RBDs. The molecular surface of ACE2 is shown in transparent pale green. The light chain of N3-1 heavily clashes with ACE2. The ACE2-binding site on RBD-down is completely blocked by H-CDR2, H-CDR3 and L-CDR3 (**e**).

Arg100a from H-CDR3, strengthening this primary Fab binding interface on RBD-up (Fig. 3c). Furthermore, the sidechain guanidinium of Arg408 on RBD-up has polar interactions with Tyr49 and Glu55 from L-CDR2, which along with H-CDR1, constitutes the secondary binding interface on RBD-up (Fig. 3c).

In contrast, the N3-1 binding site on RBD-down overlaps with the ACE2-binding site, where 11 of 19 N3-1 epitope residues are also involved in ACE2 binding. Phe486 on RBD-down inserts into a hydrophobic pocket formed by Trp96 (L-CDR3), Tyr52, Tyr58 (H-CDR2), Arg100e and Val100g (H-CDR3) (Fig. 3e). The sidechain guanidinium of Arg100e not only forms a hydrogen bond with Tyr489, but also contacts Phe486 through cation-π interactions (Fig. 3e). In addition, Asn487 and Gln 493 on RBD-down are expected to form polar interactions with Ser93 of L-CDR3 and Thr57 of H-CDR2, respectively (Fig. 3e). The angle of approach of N3-1 prevents ACE2 binding by both trapping one RBD in the 'down' position, thereby preventing exposure of the ACE2 binding site, as well as by sterically inhibiting ACE2 access to the bound 'up' RBD. (Fig. 3f). Collectively, the N3-1 antibody engages an extensive quaternary epitope on neighboring RBDs through a binding modality.

The importance of this conformational epitope engagement is made clear by comparing the binding behaviors of N3-1 IgG and N3-1 Fab, where the Fab shows a nearly 1000x fold reduction in $K_d$ against WHU1 and earlier VOC spikes and a total loss of

binding against BA.1 (Fig. 4a, Supplementary Figs. 10, 11). In authentic virus neutralization assays, N3-1 Fab was not able to neutralize WHU1 SARS-CoV-2 (Supplementary Fig. 12), further suggesting a role for the conformational epitope in neutralizing the virus and improved avidity of single IgG to the two RBD-up and one RBD-down state.

**Structure-based rational design of N3-1 partially rescues BA.1 neutralization.** The unique binding mode of N3-1 allowed it to maintain potent neutralization to all variants of concern prior to the emergence of BA.1 (Supplementary Fig. 10). Of the 15 RBD mutations in BA.1, only Q493R affects a residue making direct contact with N3-1 through its H-CDR2, residue Thr57. However, rationally designed amino acid substitutions of Thr57 failed to restore N3-1 binding potency to BA.1 in mammalian display experiments and did not markedly affect WHU1 spike binding ability (Supplementary Fig. 13), further supporting a minor role for Gln493's contribution in N3-1 epitope recognition[30].

To investigate how BA.1 escapes N3-1 neutralization, we aligned our structure of WHU1-N3-1 to apo BA.1 and instead focused on RBD mutations in the 371–376 loop (Ser371Leu, Ser373Pro, and Ser375Phe). Ser373 and Ser375 do not directly interact with N3-1, however, the orientation of the flexible loop appears inverted in BA.1, causing the occlusion of the conserved

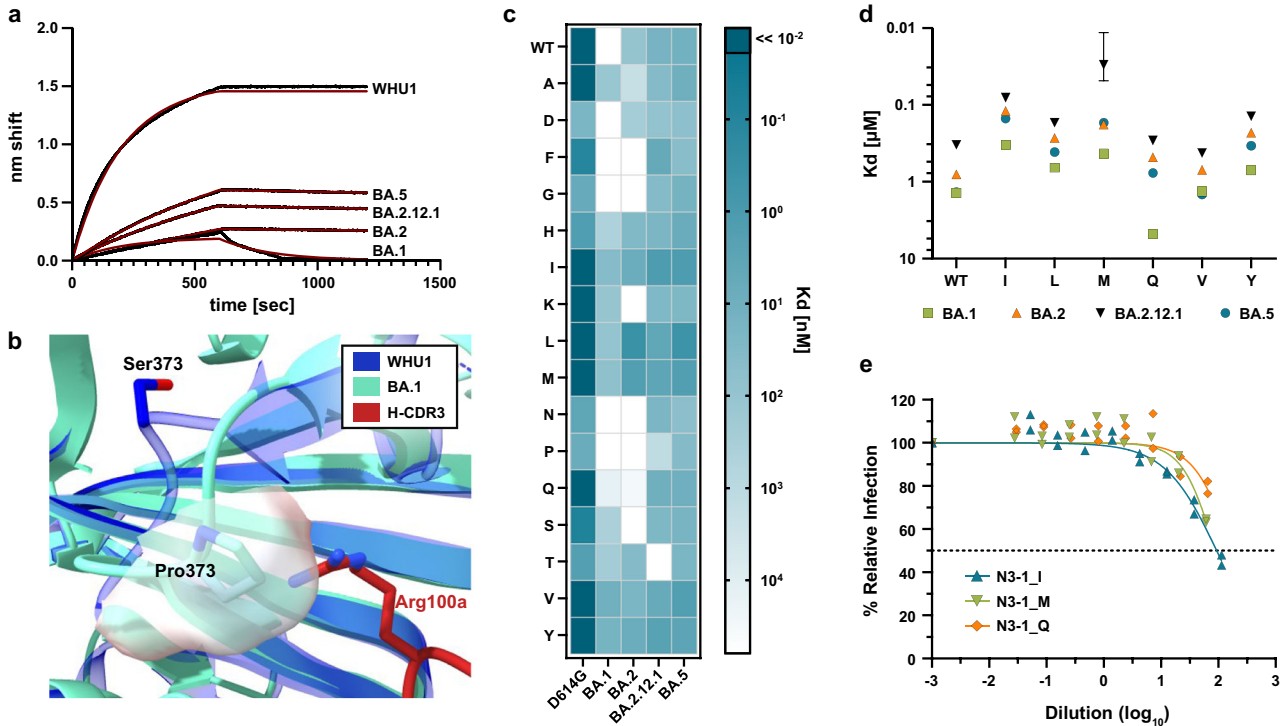

**Fig. 4 Saturation mutagenesis of Arg100a shows isoleucine substitution recovers N3-1's neutralization activity against BA.1. a** Binding and fit of N3-1 IgG to spike proteins assayed via biolayer interferometry (BLI) shows decreased affinity for Omicron relative to WHU1. **b** The S373P mutation conserved across Omicron variants occludes HCDR3 Arg100a's binding pocket, causing steric hindrance in N3-1's approach. **c** Screening of immobilized N3-1 Arg100a mutants binding to HexaPro-D614G and Omicron spike protein ectodomains by BLI. **d** Apparent binding affinity of N3-1 mutant candidates to immobilized Omicron spike variants by BLI. Bars show s.e. for the one measurement with perceptible uncertainty. **e** Virus neutralization of BA.1 by N3-1 mutants, fit to two biological replicates.

hydrophobic pocket that H-CDR3 residues Tyr98, Phe99 and Arg100a pack against (Fig. 4b). While the residues forming the hydrophobic binding pocket are largely unaffected, Pro373 now sterically hinders the approach of N3-1's H-CDR3 and specifically clashes with the guanidium moiety of Arg100a. Biolayer interferometry (BLI) measurements of N3-1 binding to WHU1 and the BA.1, BA.2, BA.2.12.1 and BA.5 variants reflect the decreased affinity to the Omicron spike proteins. The loss of binding is greatest with BA.1 despite the additional mutations accumulated in BA.5 and marked by a much faster off rate than is observed for BA.2, BA.2.12.1 and BA.5.

To determine whether binding could be rescued by substituting Arg100a, we conducted a saturation mutagenesis scan (omitting cysteine) and tested the binding of the variants against WHU1 and Omicron spikes (Fig. 4c). We identified variants I, L, M, Q, V and Y as candidates warranting further testing. All candidate variant mAbs, except for N3-1_Q, show improvements in BA.1 binding compared to WT N3-1. N3-1_I had a 4-fold improvement in $K_d$ compared to the wildtype, and against BA.2.12, N3-1_I improved WT $K_d$ by 20-fold (Fig. 4d, Supplementary Tables 3). Double mutant screening at positions 57 and 100a identified 42 variants with increased affinity to BA.1 with respect to WT N3-1 (Supplementary Table 4). Interestingly, while all variants displayed the weakest affinity towards BA.1, some neutralization was recovered against BA.1 but not against BA.2 or BA.5 (Fig. 4c, Supplementary Fig. 14).

Congruent with the $K_d$ measurements, N3-1_I was the most effective against BA.1 SARS-CoV-2 in authentic virus neutralization assays, reducing infected foci to 45.6% of untreated control (Fig. 4e). While N3-1_I addresses the steric hindrance caused by Ser373Pro, it is not able to fully restore the highly potent binding mode of N3-1 to WHU1 spike for BA.1. This is likely in part due

to the altered RBD conformation dynamics of BA.1, which more frequently adopts a two RBD-down state[16]. WHU1 samples the two RBD-up conformation more frequently, which is recognized by N3-1 and, as mentioned above, the hydrophobic binding pocket occupied by Tyr98, Phe99 and Arg100a is largely inaccessible in the RBD-down state.

## Discussion

We performed IgSeq proteomic analyses on two patient samples to identify heavy chains of serum antibodies specific to SARS-CoV-2 spike. We then adopted two strategies for light chain discovery: a rapid screening method utilizing PLCs and a YSD light chain selection. The PLC screening method, useful for quickly testing VH functionality, yielded many high-affinity antibodies. Using much larger donor light chain libraries, the YSD selections recapitulated many of same the PLC-derived VH-VL pairs and returned additional potent mAbs. In parallel to IgSeq VH discovery, we performed YSD on VH-VL libraries assembled from donor B-cell receptor repertoires. Ultimately, we found that each method independently yielded multiple neutralizing antibodies, with several of the most potent mAbs converging between approaches. By utilizing both IgSeq and YSD, we queried the serological and cellular repertoires, respectively. Clonal diversity among the antigen-specific peripheral B-cells is distinct from that of the circulating serum antibodies, and mining from both repertoires provides better understanding of an individual's immune response and gives more opportunity to identify strong neutralizing mAbs[31].

The cryo-EM structure of N3-1 is of particular importance because it reveals a binding mode at a quaternary epitope of the spike trimer, allowing this antibody to bind the RBD in both the

'up' and 'down' conformations. This enables binding to all three RBD subunits in the spike protein trimer simultaneously and greatly enhances the binding affinity of the full mAb. N3-1 binding requires a spike trimer with two RBDs in the 'up' and one in the 'down' conformation. Therefore, N3-1 association and dissociation are vulnerable to altered RBD dynamics in which the up position is sampled less frequently compared to the Wuhan-Hu-1 spike.

The predominant configuration of the Omicron S-protein alternates between one RBD up, two RBDs down or all three RBDs down[12]. This largely explains the attenuated binding of N3-1 to Omicron variants and shows how altered RBD dynamics present an additional mechanism of immune evasion. This is supported by our recent findings that reversion of single Omicron RBD mutations to Wuhan-Hu-1 alone does not fully restore binding of the quaternary epitope[32], results congruent with our observation that engineered N3-1 (N3-1_I/M) only partially compensates for the loss of affinity to BA.1.

N3-1 interacts with a conserved hydrophobic pocket, but because Omicron spikes preferentially adopt altered RBD equilibria, the neutralizing quaternary epitope is less available. This is made apparent in that BA.1 requires additional mutations in the S2 to fully escape N3-1. In contrast, BA.2 RBD mutations that further stabilize the RBD-down conformation are sufficient for escape of N3-1[32]. Despite improved binding kinetics to BA.2, BA.2.12.1, and BA.5 spikes by rational design of N3-1 Arg100e mutants, these enhanced binders fail to neutralize Omicron variants—likely due to constrained RBD dynamics and interactions.

Understanding the relevance of this immune evasion mechanism presents new opportunities for improved vaccine design. While WHU1 readily adopts the two RBD 'up', one RBD 'down' state, it stands to reason that it nevertheless transiently adopts two RBD 'down', one RBD 'up' or all RBD 'down' conformations. By creating stabilized constructs of alternate spike conformational states as part of strain-specific or multivalent vaccines, the immune system would then be exposed to epitopes, which would otherwise be subdominant, evoking a more broadly protective antibody response. Thus, engineering immunogens with multiple conformational states may lead to more robust vaccines that are less susceptible to common escape mutations and provide greater protection potential.

## Methods

**Strains and media**. E. coli strain DH10B was used for all routine cloning of yeast and mammalian constructs. Yeast strain EBY100 (MATa AGA1::GAL1-AGA1::URA3 ura3-52 trp1 leu2-delta200 his3-delta200 pep4::HIS3 prbd1.6 R can1 GAL) was acquired from ATCC (cat. no. MYA-4941) and used for antibody surface display and selection. To improve antibody surface expression the human chaperone BIP (binding immunoglobulin protein) and yeast PDI (protein disulfide isomerase) were genomically integrated as an expression cassette in the HO locus. Yeast were grown in rich medium (YPD; Takara, cat. no. 630409) or in the selective medium for leucine prototrophs after library transformation (Takara cat. no. 630310). YEP-galactose was used for the expression of displayed antibody libraries (1% yeast extract, 1% bacto-peptone, 0.5% NaCl, 2% galactose, 0.2% glucose).

**Antigens and antibodies**. Stabilized spike ECD antigen S-2P was biotinylated using the EZ-link kit (Thermo Scientific, cat. no. 21435) and labeled with streptavidin-AF647 (Invitrogen, cat. no. S32357). RBD was labeled with a mouse anti-human Fc-AF647 (Southern Biotech, cat. no. 9042-31). Fab library light chains were labeled with anti-FLAG M2-FITC (Sigma, cat. no. F4049).

**Donors**. The acquisition of blood specimens from convalescent individuals was approved by the University of Texas at Austin Institutional Review Board (protocol 2020-03-085; Breadth of serum antibody immune responses prior to, or following, patient recovery in asymptomatic and non-severe COVID-19). Informed consent was obtained from both participants. Blood was collected from two PCR-confirmed patients at day 12 post-onset of symptoms. Neither donor was hospitalized or experienced severe disease. PBMCs and plasma were both collected by density gradient centrifugation using Histopaque-1077 (Sigma-Aldrich).

**Preparation of serum antibodies for IgSeq proteomic analysis**. Total IgG was isolated from patient serum using Pierce Protein G Plus Agarose (Pierce Thermo Fisher Scientific) and cleaved into F(ab')2 fragments with IdeS protease. Antigen-specific F(ab')2 was enriched by affinity chromatography against recombinant SARS-CoV-2 S-2P or RBD protein cross-linked to NHS-activated agarose resin (Thermo Fisher Scientific). Eluted F(ab')2 fractions were concentrated by vacuum centrifugation and resuspended in a Digestion Buffer (50 mM Tris, pH 7.8, 2 mM calcium chloride). Samples were mixed with an equal volume of 2,2,2 tri-fluoroethanol (TFE) and reduced with a final concentration of 5 mM TCEP at 56 °C for 45 minutes, followed by alkylation with 25 mM iodoacetamide at 25 °C. Samples were diluted 10-fold with Digestion Buffer and 1 μg trypsin was added. Proteins were digested for 4 hours at 37 °C. The digestion was halted with the addition of formic acid to 1% (v/v), and peptides were bound, washed, and eluted from Hypersep C18 SpinTip columns (Thermo Scientific) according to the manufacturer's protocol. Eluted peptides were dried by vacuum centrifuge and resuspended in 5% acetonitrile, 0.1% formic acid.

**LC-MS/MS analysis of antigen-enriched antibodies**. Liquid chromatography-tandem mass spectrometry analysis was carried out on a Dionex Ultimate 3000 RSLCnano system coupled to an Orbitrap Fusion Lumos Mass Spectrometer (Thermo Scientific). Samples were loaded onto an Acclaim PepMap 100 trap column (75 μm × 2 cm; Thermo Scientific) and separated on an Acclaim PepMap RSLC C18 column (75 μm × 25 cm; Thermo Scientific) with a 3%-40% acetonitrile gradient over 60 min at a flow-rate of 300 nl/min. Peptides were eluted directly into the Lumos mass spectrometer using a nano-electrospray source. Mass spectra were acquired in data-dependent mode with a 3 sec. cycle time. Full (MS1) scans were collected by FTMS at 120,000 resolution (375-1600 m/z, AGC target = 5E5). Parent ions with a positive charge state of 2-6 and minimum intensity of 3.4E4 were isolated by quadrupole (1 m/z isolation window) and fragmented by HCD (stepped collision energy = 30 +/−3%). Fragmentation (MS2) scans collected by ITMS (rapid scan rate, AGC target = 1E4). Selected ions and related isotopes were dynamically excluded for 20 sec (mass tolerance = +/−10ppm).

**Antibody variable chain sequencing**. Peripheral blood mononuclear cells (PBMCs) from processed donor samples were provided in Trizol as a kind gift from Dr. Gregory C. Ippolito. IgG and IgM VH cDNA libraries were separately amplified from PBMC RNA of two donors and sequenced to create donor-specific reference databases, from which the complete amino acid sequences of serum IgG proteins could subsequently be determined based on their mass spectral identifications. Supplementary Fig. 1 shows the distribution and diversity of clonotypes and V-gene usage for the variable heavy chain repertoires of donors 1 and 2.

**IgSeq MS data analysis**. Mass spectra were analyzed using Proteome Discoverer 2.2 software (Thermo Scientific). Precursor masses were first recalibrated with the Spectrum File RC node using a consensus human reference proteome database (UniProt) with common contaminants (MaxQuant) and precursor mass tolerance of 20 ppm. Recalibrated mass spectra were searched against a custom database for each donor consisting of donor-derived VH sequences, VL sequences, and the human and contaminant sequences using the Sequest HT node. Mass tolerances of 5 ppm (precursor) and 0.6 Da (fragment) were used. Static carbamidomethylation of cysteine ($+57.021$ Da) and dynamic oxidation of methionine ($+15.995$ Da) were considered. False discovery rates for peptide-spectrum matches (PSMs) were estimated by decoy-based error modelling through the Percolator node. Label-free quantitation (LFQ) abundances were calculated from precursor areas using the Minora Feature Detector and Precursor Ions Quantifier nodes.

Resulting PSMs were filtered according to methods described (Boutz, 2014). Briefly, peptide sequences differing only by isoleucine/leucine substitution were considered equivalent and combined into a single PSM. PSMs were re-ranked by posterior error probability, q-value, and Xscore. Only top-ranked, high-confidence PSMs (FDR < 1%) were retained for each scan. If two or more PSMs had identical top-ranked scores, they were considered ambiguous and removed. PSMs for the same peptide sequence were summed and the average mass deviation (AMD) was calculated for each peptide. Peptides with AMD greater than 2 ppm were filtered out. Peptides mapping to VH sequences from a single clono-group were considered clono-specific. Clono-specific peptides overlapping the CDR3 sequence by four amino acids or more were considered CDR3-informative.

For each clono-group, PSMs and LFQ abundances of clono-specific CDR3-informative peptides were summed. Ratios of elution:flow-through PSMs and LFQ abundances were calculated; only clono-groups with both ratios > 5 were considered elution-specific.

**Public light chain screening**. We analyzed a previously published dataset of VH-VL pairs and chose the top 6 most abundant kappa and top 3 most abundant lambda V-genes (Fig. 2a). We then constructed germline versions of these V-genes with their most frequently observed J gene as published on the iRepertoire website[20–23] (table S1) and obtained full-length light chains for each PLC. SARS-CoV-2 specific IgSeq or YSD heavy chains were then expressed with each PLC as a full-length IgG1 and screened for binding through ELISA using recombinant S2P.

**Library assembly and bacterial transformation**. Donor B-cell VH and VL amplicons were amplified via PCR to include adapters for cloning into yeast expression vectors. Assembly into the yeast kappa and lambda expression vectors was done via Golden Gate assembly. Library assemblies were prepared in 20 μL reactions as follows: 2 μL 10X AARI buffer (ThermoFisher Scientific, cat. no. B27), 0.4 μL 50X oligo buffer (ThermoFisher Scientific, cat. no. ER1582), 0.2 μL 100 mM ATP (ThermoFisher Scientific, cat. no. R0441), 20 fmol backbone DNA, 40 fmol VH and VL amplicons, 0.5 μL (2 U/μL) AARI endonuclease (ThermoFisher Scientific, cat. no. ER1582), and 0.5 μL T7 ligase (3000 U/μL) (NEB, cat. no. M0318). Each assembly was scaled up to 16 total reactions in 8-well strips. Thermocycling consisted of the following protocol: 37°C, 15 minutes; 37 °C, 2 minutes, 16 °C, 1 minute; go to step 2, x74; 37 °C, 60 minutes; 80°C, 15 minutes; hold at 4 °C. Assemblies were consolidated and column purified using Promega Binding Solution (Promega, cat. no. A9303) to bind DNA to a Zymo-spin II column (Zymo Research, cat. no.

C1008). The column was washed twice with DNA Wash Buffer (Zymo Research, cat. no. D4003) and eluted in 30 μL nuclease-free water. For library transformations, DH10B E. coli cells were diluted 1:100 from confluent culture into 50 mL Superior broth (AthenaES, cat. no. 0105). When cells reached an $OD_{600}$ of 0.4–0.6, they were washed 3X with cold 10% glycerol and resuspended to a final volume of 600 μL. The purified library was added to cells and electroporated at 2.5 kV in an E. coli Pulser electroporator (Bio-Rad) using Genepulser 0.2 cm cuvettes (Bio-Rad, cat. no. 1652086) at 200 μL per transformation.

**Library transformation into yeast and protein expression**. Purified libraries were linearized for integration into the yeast genome via homologous recombination at the Leu2 locus. For each 1 μg library plasmid, 0.5 μL NotI (10 units/μL) (NEB, cat. no. R0189) was used with the supplied Buffer 3.1 in 10 μL. Reactions were incubated at 37 °C overnight and heat-inactivated at 80 °C for 20 minutes. Digests were pooled and column purified as described in previous sections and eluted in 25 μL nuclease-free water. Our strain was electroporated using 10 μg or 20 μg of linearized DNA. We found that 10 μg of linearized DNA was sufficient for library sizes of $10^6$, and that library sizes could reach $>10^7$ with 20 μg DNA. Transformed yeast were recovered in SD-Leu medium (see "strains and media" section). Libraries were passaged once at 1:100 before protein expression to reduce contamination from untransformed cells. To express Fab libraries, yeast were washed in YEP-galactose (see "strains and media") and diluted 1:10 into 10 mL final volume. Cells were induced for 48 hours at 20 °C with shaking.

**Fab library labeling and selection**. Expressed yeast libraries were harvested at 100 μL (representing approximately $10^7$ cells) and washed with PBSA buffer (1X PBS, 2 mM EDTA, 0.1% Tween-20, 1% BSA, pH 7.4). Antigen was incubated with cells in 1 mL in PBSA at 200 nM at RT for one hour, washed with PBSA at 4 C, and labeled with secondary antibodies (mouse anti-human FITC, 1:100; streptavidin-AF647, 1:100; mouse anti-human Fc-AF647, 1:50). Cells were washed 2X and resuspended in 2 mL in cold PBSA for sorting. Cell sorting was performed using a Sony SH800 fluorescent cell sorter. For first-round libraries, cells were sorted into 2 mL SD-Leu medium supplemented with penicillin/streptomycin (Gibco, cat. no. 15140122). Cells were recovered by shaking incubation for 1-2 days for further rounds of selection or plated directly for phenotyping clones.

**Phenotyping assays**. Sorted clones from rounds two or three of selection were picked into microplates. After antibody expression, 10 μL of cells were transferred to a fresh 2-mL microplate and washed 2X with 200 μL cold PBSA buffer. Cells were labeled with 200 nM spike or RBD antigen in 50-100 μL PBSA at RT for 1 h with shaking at 1000 rpm, 3 mm orbit. Labeled cells were washed 2X, and secondary labels were applied for 25 minutes at 4 C and washed twice. Cells were resuspended in 200 μL ice-cold PBSA just before analysis. Samples were analyzed on a Sony SA3800 Spectral Cell Analyzer.

**Next generation sequencing**. Genome extraction was performed on yeast cultures of libraries and sorted rounds underwent genome extraction using a commercial kit (Promega, cat. no. A1120) with zymolyase (Zymo Research, cat. no. E1004). 100 ng genomic template was used to amplify the heavy and light chains separately or as one amplicon for short or long-read sequencing, respectively. For amplification of heavy chain genes only, primers JG.VHVLK.F and JG.VH.R were used. For amplification of light chain genes only, primers JG.VL.F and JG.VHVK.R or

JG.VHVL.R were used for kappa and lambda vectors, respectively. For amplification of paired genes, primers JG.VHVLK.F and JG.VHVK.R or JG.VHVL.R were used. Amplicons were column purified and deep sequenced with an iSeq. In parallel, we obtained ~1.8 kb sequences spanning the entire VH and VL using MinION nanopore sequencing (Oxford Nanopore Technologies Ltd., MinION R10.3).

**Colony PCR and Sanger sequencing**. Sorted yeast populations were plated on SD -Leu and 8-32 colonies per plate were picked into 2 mL microplates either by hand or using a QPIX 420 (Molecular Devices) automatic colony picker. Cultures were grown at 1000 rpm at 3 mm orbit at 30°C overnight. Cells (20 μL) were transferred to a fresh microplate and washed with 1 mL TE buffer (10 mM Tris, 1 mM EDTA). Cells were incubated with 20 μL zymolyase solution (5 mg/mL zymolyase, 100 T in TE) at 37°C for 1 hour. Cells (5 μL) were then used in colony PCR to amplify the paired heavy and light chains. Amplicons were column purified with the Wizard SV 96 PCR Clean-Up System (Promega, cat. no. A9342) and yields were quantified with a Nanodrop spectrophotometer or the Quant-it Broad-Range dsDNA kit (Invitrogen, cat. no. Q33130). Approximately 10 ng (2.5-5 μL) of purified PCR products were then subjected to Sanger sequencing.

**Long-read sequencing (donors 1 & 2)**. Sequencing libraries were prepared from 18 amplicon samples using the Native Barcoding Kit (Oxford Nanopore Technologies; cat. no. EXP-NBD103) paired with the Ligation Sequencing Kit (Oxford Nanopore Technologies; cat. no. SQK-LSK109) according to the manufacturer's directions. Between four and eight sequencing libraries per flow cell were pooled for sequencing on three MinION flow cells (Oxford Nanopore Technologies; R9.4.1) for 72 hours on an Oxford Nanopore Technologies MinION Mk1B device (Oxford Nanopore Technologies). Raw data was basecalled using the high-accuracy model in Guppy (v.3.5.2).

**Short-read sequencing (phenotyping plates)**. Sequencing libraries were prepared from 308 amplicon samples using the Nextera DNA Flex Library Preparation kit (Illumina; cat. no. 20018705) according to the manufacturer's directions. Sequencing libraries were pooled and sequenced (2x151bp) on an iSeq 100 (Illumina; California, USA) using iSeq 100 i1 Reagents v.1 (Illumina; cat. no. 20021533).

**Long-read sequencing (IgSeq-YSD)**. Sequencing libraries were prepared from 16 amplicon samples using the Native Barcoding Kit (Oxford Nanopore Technologies; Cat. No. EXP-NBD104) paired with the Ligation Sequencing Kit (Oxford Nanopore Technologies; Cat. No. SQK-LSK109) according to the manufacturer's directions. Four sequencing libraries were pooled per flow cell and sequenced on four GridION flow cells (Oxford Nanopore Technologies; R9.4.1) for 72 hours on a GridION Mk1 device (Oxford Nanopore Technologies; Oxford, England, UK). Raw data was live basecalled using the high accuracy model in Guppy (v.4.0.11).

**Short-read sequencing (IgSeq-YSD)**. Sequencing libraries were prepared from 16 samples using the Nextera DNA Flex Library Preparation kit (Illumina; cat. no. 20018705) according to the manufacturer's directions. Sequencing libraries were pooled and sequenced (2×150 bp) on an iSeq 100 (Illumina; California, USA) using the iSeq 100 i1 Reagents v2 (Illumina; Cat. No. 2009584).

**Sequence processing and consolidation into VHVL clones**. Individual reads after Guppy base calling typically average more than 10% error per base, and numerous tools exist to align and reduce reads into consensus sequences with substantially improved accuracy. However, such tools were not designed for antibody library sequencing and its huge populations of subtly different sequences which, even assuming successful alignment, group into a myriad of very short, disconnected assemblies. We therefore implemented a bioinformatic pipeline to obtain accurate VHVL sequences from the MinION and iSeq data and to estimate, within each YSD round, the relative abundance of individual VHVL pairs (Supplementary Fig. 6).

Our methods proceed through antibody V(D)J annotation of raw MinION reads using MiXCR (v3.0.13)[33]; iteratively growing and shrinking sequence clusters based on annotated features from each read; sequence error correction and consolidation within each cluster, optionally including high-quality Illumina reads; and finally, VHVL clone definition within each sample and quantitation by number of reads mapped to each clone. For enumeration, we only include counts for reads with a length of 1700 to 2100 base pairs. This points to a secondary advantage of MinION sequencing, as shorter reads proliferate during PCR and inflate apparent abundance of particular species. Without length-filtering, relative VHVL abundance calculated from both MinION and iSeq reads are strongly correlated (Supplementary Fig. 7).

**Tissue culture and transient transfection of WA1-S2P ECD and RBD**. Spike ECD protein and RBD proteins were expressed in Expi293F cells (Thermo Fisher; cat. no. A14527) using the manufacturer-provided guidelines with slight modifications. Cells were transferred to a 125 mL non-baffled vented shake flask containing 29 mL of fresh pre-warmed ExpiExpression medium at 37°C. Cells were incubated in at 37°C with ≥80% humidity and 8% $CO_2$ on an orbital shaker at 120 rpm and grown until they reached a cell density of $3 \times 10^6$ viable cells/ml. Fresh pre-warmed ExpiExpression medium was added to 1 L non-baffled vented shake flask and the 30 mL cell suspension was carefully introduced, making the seeding density of $0.4 \times 10^6$ viable cells/mL in a final culture volume of 225 mL. After the cell density reached $3 \times 10^6$ viable cells/ml, the culture was expanded to a final volume of 2.25 L in two 2.8 L Thomson Optimum Growth flasks with a seeding density of $0.3 \times 10^6$ viable cells/mL. After the cell density reached $3 \times 10^6$ viable cells/mL, 2 L of the culture was split into four Thompson flasks with each flask containing 500 mL of culture medium and 500 mL of fresh pre-warmed ExpiExpression medium. The final culture volume in each of the four flasks was 1 L. The cells were incubated at 37°C with ≥80% humidity and 8% $CO_2$ on an orbital shaker at 100 rpm. When the cells reached a density of $3 \times 10^6$ viable cells/mL, the culture was transferred to two 500 mL sterile centrifuge bottles and the cells were spun down at 100 x g for 10 min. The supernatant was removed, and the cells were resuspended in 4 L of fresh, prewarmed medium. The cells were allowed to re-stabilize in the incubator for 24 h and were transfected at $3 \times 10^6$ viable cells/mL in 4 L.

Transfection was performed by diluting 4 mg of plasmid DNA (pDNA) in 240 mL of OptiMEM medium in a sterile bottle and gently inverting 3 − 4 times before incubating at room temperature (RT) for 5 min. ExpiFectamine 293 reagent (13 mL) was then diluted in 225 mL OptiMEM in a sterile bottle and inverted 3−4 times before incubating for 5 min at RT. The diluted pDNA and ExpiFectamine reagent were carefully mixed and incubated at RT for 15 min. One-fourth of the combined complex was then slowly transferred to each of the flasks while gently swirling the cells during addition. The cultures were again placed in the incubator with shaking at 100 rpm for 18 h. Post-

transfection, ExpiFectamine 293 Transfection Enhancer 1 (6 mL) and ExpiFectamine 293 Transfection Enhancer 2 (60 mL) were added to each flask. The cell viability was monitored every 24 h and the cells were harvested when the viability dropped below 70% or after approximately 3 d. Harvesting was done by centrifugation at 15,900 x g for 45 min at 4°C and the supernatant was transferred into sterile bottles.

**Antigen purification**. The feed was prepared adding 0.2 M NaCl and 10 mM imidazole to the supernatant while mixing. The feed material was filtered using a 0.45 µm PES filter membrane pre-wetted with PBS before loading on a prepared Ni-IMAC column. A metal affinity column was prepared by packing IMAC FF beads (Cytiva) into an AxiChrom 70-column housing to a bed height of 9.5 cm and then charging with 50% column volume (CV) of a 0.2 M nickel sulfate solution, washing with water and then 50% CV of 100% "B" buffer (50 mM sodium phosphate buffer containing 300 mM NaCl and 250 mM imidazole, pH 7.8) to remove weakly bound nickel ions. The column was then washed with 100% "A" buffer (50 mM sodium phosphate buffer containing 300 mM NaCl and 20 mM imidazole, pH 7.2) prior to sample loading. The prepared (4.8 L) feed was then loaded onto the column at a linear flow rate of 90 cm/h. After loading, the column was washed with two CVs of 100% A buffer and two CVs of a 13% B buffer (containing 50 mM imidazole) before eluting the protein using 100% B (250 mM imidazole) for 3 CVs. All steps except for the loading step were done at 150 cm/h.

Fractions from the elution peak were pooled and then concentrated 4- to 8-fold by ultrafiltration (UF) using a 115 cm$^2$ hollow fiber cartridge with either a 50 kDa (S-2P) or a 10 kDa (RBD) molecular weight cutoff membrane (Repligen) and then diafiltered after concentration by exchanging with 5 volumes of PBS.

The concentration of diafiltered protein was determined by measuring the absorbance at 280 nm versus a PBS blank. The protein concentration in mg/mL was obtained using a divisor of 1.03 mL/mg-cm for S-2P and 1.19 mL/mg-cm for RBD. A qualitative assessment of protein quality was made using SDS-PAGE with SYPRO Ruby staining (BioRad) for reduced and non-reduced samples. Only those preparations showing predominantly full-length S-2P (160 kDa subunits) or RBD (70 kDa) were used in ELISAs for assessing antibody binding.

**Monoclonal antibody expression and purification**. VHVL candidates were cloned into custom Golden Gate-compatible pCDNA3.4 vectors for IgG1 expression. For transfections, VL was mixed 3:1 with a corresponding VH. Plasmids were transfected into Expi293F (Invitrogen) cells using the recommended protocol. Monoclonal antibodies were harvested at 5-7 days post-transfection. Expi293F cells were centrifuged at 300x *g* for 5 min, supernatants were collected and centrifuged at 3000x *g* for 20 min at 4°Cand diluted to 1X PBS final concentration. Each supernatant was passed through a Protein G or A agarose affinity column (Thermo Scientific). Flow through was collected and passed through the column three times. Columns were washed with 10 CV of PBS and antibodies were eluted with 5 mL 100 mM glycine, pH 2.5 directly in neutralization buffer containing 500 µL 1 M Tris-HCl, pH 8.0.

**Analysis of mammalian cell surface displayed spike proteins with flow cytometry**. The assay has been described in detail in the recent paper by Javanmardi et al[29]. Briefly, plasmids expressing full-length SARS-CoV-2 spike protein variants and WT (HexaPro-D614G) as Spike-Linker-3XFLAG-TM were transfected into HEK293T cells (ATCC CRL- 3216) using

Lipofectamine 2000 according to manufacturer's instructions. After 48 h, cells were collected and resuspended in PBS-BSA and incubated with anti-FLAG (mouse) and anti-spike (human) mAbs for 1 h, shaking at RT. Cells were washed 3X with PBS-BSA, resuspended in 1 mL and incubated with Alexa Fluor 488 (anti-mouse) and Alexa Fluor 647 (anti-human) antibodies for 30 min, shaking at 4°C. Cells were washed again and resuspended in PBS-BSA prior to flow cytometry analysis (SA3900 Spectral Analyzer, Sony Biotechnology). All data was analyzed with FlowJo (BD Bioscience).

**ELISA**. Antigen ELISA plates were made using high-binding plates (Corning, cat. no. 3366) with antigen diluted in PBS to a final concentration of 2 µg/mL. Antigen solution (50 µL) was added to microplates and incubated overnight at 4°C with shaking at 100 rpm, 3 mm orbit. Plates were blocked with PBSM (2% milk in PBS) at RT for 1 h. Plates were washed 3X with 300 µL PBS-T (0.1% Tween-20). Purified antibodies were prepared to 10 µg/mL in PBSM and serially diluted. Antibodies were incubated for 1 h at RT. Plates were washed 3X with PBS-T, and secondary goat anti-human Fab-HRP (Sigma-Aldrich, cat. no. A0293) was applied at 1:5000 in PBSM in 50 µL and incubated at RT for 45 min. HRP substrate (50 µL) was added to wells and the reaction proceeded for 5-15 min until quenched with 50 µL 4 M H$_2$SO$_4$ and analyzed for absorbance at 450 nm in a plate reader.

**BLI**. All N3-1 variant BLI measurements were obtained on a GatorPlus from GatorBio. Purified antibodies and spike variants were buffer exchanged into PBS and diluted in K-Buffer (GatorBio Cat#120011). For initial $K_d$ measurements antibodies were loaded onto anti-human IgG FC (HFC) Gen II probes (GatorBio Cat#1600024) and dipped into spike variants at 200 nM concentration for 600 seconds followed by an equal dissociation time. For more precise characterization of N3-1 variants of interest, the antibodies were serially diluted 1:3 starting at 750 ug/ml. The omicron spike variants were loaded onto Streptactin XT probes (GatorBio) and dipped into the antibody dilutions and allowed to associate for 600 seconds followed by a 600 second dissociation period. Probes were regenerated between assays using RegenBuffer (GatorBio Cat#120012) and assays performed in 384 well polypropylene plates (Costar #3658). Bind fitting and kinetic analysis was performed with the GatorOne software version 2.10.0713. Supernatant screening was carried out using a starting dilution of 1:1 clarified, PBS-buffered supernatants in Q buffer.

**Authentic virus neutralization assays**. Assays were performed by three independent groups. The first SARS-CoV-2 micro-neutralization assay was adapted from an assay used to study Ebola virus. This assay also used SARS-CoV-2 strain WA1. Antibodies were diluted in cell culture medium in triplicate. A SARS-CoV-2 monoclonal antibody was used as a positive control. An antibody that does not bind SARS-CoV-2 was used as negative control. Diluted antibodies were mixed with the SARS-CoV-2 WA1 strain, incubated at 37 °C for 1 h, then added to Vero-E6 cells (ATCC CRL-1586) at target MOI of 0.4. Unbound virus was removed after 1 h incubation at 37 °C, and culture medium was added. Cells were fixed 24 h post-infection, and the number of infected cells was determined using SARS-CoV-S specific mAb (Sino Biological, cat. no. 401430-R001) and fluorescently labeled secondary antibody. The percent of infected cells was determined with an Operetta high-content imaging system (PerkinElmer) and Harmony software. Percent neutralization for each monoclonal antibody at each dilution was determined relative to untreated, virus-only control wells. The second set of virus neutralization

(VN) assays for variants of concern and WA1 was performed in an orthogonal assay. The ability of the monoclonal antibodies to neutralize SARS-CoV-2 was determined with a traditional VN assay using SARS-CoV-2 strain USA-WA1/2020 (NR-52281-BEI resources). All experiments with SARS-CoV-2 were performed in the Eva J Pell BSL-3 laboratory at Penn State and were approved by the Penn State Institutional Biosafety Committee (IBC # 48625). For each mAb a series of 12 two-fold serial dilutions were assessed. Triplicate wells were used for each antibody dilution. 100 tissue culture infective dose 50 (TCID50) units of SARS-CoV-2 were added to 2-fold dilutions of the diluted mAb. After incubating for 1 hour at 37°C, the virus and mAb mixture was then added to Vero E6 cells in a 96-well microtiter plate and incubated at 37°C. After 3 days, the cells were stained for 1 hour with crystal violet–formaldehyde stain (0.013% crystal violet, 2.5% ethanol, and 10% formaldehyde in 0.01 M PBS). The end-point of the microneutralization assay was determined as the highest mAb dilution, at which all 3, or 2 of 3, wells are not protected from virus infection. Percent neutralization ability of each dilution of the mAb was calculated based on the number of wells protected, 3, 2, 1, 0 of 3 wells protected was expressed as 100%, 66.6%, 33.3%, or 0%.

For the third measure monoclonal antibody neutralization titers, we used a fluorescent focus reduction neutralization test (FFRNT) with an mNeonGreen (mNG) reporter SARS-CoV-2 (strain USA-WA1-2020) or SARS-CoV-2 (strain USA-WA1-2020) bearing a variant spike gene (Omicron BA.1). The construction of the mNG USA-WA1-2020 SARS-CoV-2 bearing variant spikes has been reported previously (Xie et al., 2021; Zou et al., 2022). For the FFRNT assay, we seeded $2.5 - 3 \times 10^4$ Vero E6 cells (ATCC CRL-1586) into black, mCLEAR flat-bottom 96-well plates (Greiner Bio-One 655096). We incubated plates at 37 C with 5% CO2 overnight. The next day, each sample was two-fold serially diluted in culture medium with an initial dilution of 1:20. We incubated diluted antibody with 100-150 fluorescent focus units (FFU) of mNG SARS-CoV-2 at 37 C for 1 h before loading the serum-virus mixtures into 96-well plates pre-seeded with Vero E6 cells. Following a 1 h infection period, we removed the inoculum and added overlay medium (100 mL DMEM + 0.8% methylcellulose, 2% FBS, and 1% penicillin/streptomycin). We then incubated the plates at 37 C for 16 h and acquired raw images of mNG fluorescent foci using a CytationTM 7 (BioTek) cell imaging reader with a 2.53 FL Zeiss objective and wide field of view. We used GFP software settings [469 nm, 525 nm], a threshold of 4000, and an object selection size of 50-1000 mm during image processing. For relative infectivity calculations, we counted and normalized the foci in each well relative to non-antibody-treated controls. We plotted curves of relative infectivity versus serum dilution using Prism 9 (GraphPad). We used a nonlinear regression method to determine the dilution fold at which 50% of mNG SARS-CoV-2 was neutralized, defined as FFRNT50 in GraphPad Prism 9. Each antibody was tested with two biological replicates.

**Surface plasmon resonance**. To investigate the binding kinetics of mAb N3-1 binding to the spikes, purified His-tagged spike variants (SARS-CoV Tor2 S-2P, SARS-CoV-2 Wuhan-Hu-1 S-HexaPro, SARS-CoV-2 D614G S-HexaPro, SARS-CoV-2 B.1.1.7 S-Hexapro, SARS-CoV-2 B.1.351 S-HexaPro and SARS-CoV-2 B.1.1.529.1/BA.1 S-HexaPro) were immobilized on a Ni-NTA sensor chip (GE Healthcare) using a Biacore X100 (GE Healthcare). For Fab binding experiments, we immobilized spike proteins to a level of ~450 response units (RUs). Serial dilutions of purified Fab N3-1 were injected at concentrations ranging from 400 to 6.25 nM over spike-immobilized flow cell

and the control flow cell in a running buffer composed of 10 mM HEPES pH 8.0, 150 mM NaCl and 0.05% Tween 20 (HBS-T). Between each cycle, the sensor chip was regenerated with 0.35 M EDTA, 50 mM NaOH and followed by 0.5 mM $NiCl_2$. For IgG binding experiments, spike immobilization of 200 RUs was used instead to avoid mass transport effect. Serial dilutions of purified IgG N3-1 were injected at concentrations ranging from 25 to 1.56 nM over a spike-immobilized flow cell and the control flow cell. For the SARS-CoV Tor2 S-2P binding experiments, IgG N3-1 concentrations ranging from 100 to 6.25 nM were used. Response curves were double-reference subtracted and fit to a 1:1 binding model or heterogeneous ligand binding model using Biacore X100 Evaluation Software (GE Healthcare).

**Negative stain EM for spike-IgG complexes**. To investigate mAb N3-1 binding to spike proteins, purified SARS-CoV-2 Wuhan-Hu-1 S-HexaPro was incubated with 1.2-fold molar excess of IgG N3-1 in 2 mM Tris pH 8.0, 200 mM NaCl and 0.02% $NaN_3$ on ice for 10 min. The spike-IgG complexes were at a concentration of 0.05 mg/mL in 2 mM Tris pH 8.0, 200 mM NaCl and 0.02% $NaN_3$ prior to deposition on a CF-400-CU grid (Electron Microscopy Sciences) that was plasma cleaned for 30 sec in a Solarus 950 plasma cleaner (Gatan) with a 4:1 ratio of $O_2/H_2$ and stained using methylamine tungstate (Nanoprobes). Grids were imaged at a magnification of 92,000X (corresponding to a calibrated pixel size of 1.63 Å/pix) in a Talos F200C TEM microscope equipped with a Ceta 16 M detector. The CTF-estimation, particle picking and 2D classification were all performed in cisTEM[34].

**Cryo-EM sample preparation and data collection**. Purified SARS-CoV-2 Wuhan-Hu-1 S-HexaPro at 0.2 mg/mL was incubated with 5-fold molar excess of Fab N3-1 in 2 mM Tris pH 8.0, 200 mM NaCl and 0.02% $NaN_3$ at RT for 30 min. The sample was then deposited on plasma-cleaned UltrAuFoil 1.2/1.3 grids before being blotted for 4 sec with −3 force in a Vitrobot Mark IV and plunge-frozen into liquid ethane. For the HexaPro-N3-1 sample, 3,203 micrographs were collected from a single grid. FEI Titan Krios equipped with a K3 direct electron detector (Gatan) was used for imaging. Data were collected at a magnification of 22,500x, corresponding to a calibrated pixel size of 1.07 Å/pix. A full description of the data collection parameters can be found in Table 1.

**Cryo-EM data processing**. Gain reference- and motion-corrected micrographs processed by Warp[35] were imported into cryoSPARC v2.15.0[36], which was used to perform CTF correction, micrograph curation, particle picking, and particle curation via iterative rounds of 2D classification. The final global reconstructions were then obtained via ab initio reconstruction, iterative rounds of heterogeneous refinement, and subsequently nonuniform homogeneous refinement of final classes with C1 symmetry. To better resolve the Fab-spike interfaces, the dataset was subjected to particle subtraction and focused refinement. Finally, global and focused maps were sharpened using DeepEMhancer[37]. For N3-1-RBDs model building, we used one RBD-up and one RBD-down from S-HexaPro (PDB ID: 6XKL[38]) and two homologous Fab structures (PDB ID: 5BV7 and 5ITB) as an initial model to build into map density via UCSF ChimeraX[39]. The model was built further and iteratively refined using a combination of Coot[40], Phenix[41], and ISOLDE[42]. The detailed workflows of cryo-EM data processing and data validation can be found in Supplementary Figs. 15-16 and Table 1. We used PISA to calculate the hydrogen bonds, salt

**Table 1 N3-1 Cryo-EM Statistics.**

| EM data collection | | | |
|---|---|---|---|
| Microscope | FEI Titan Krios | | |
| Voltage (kV) | 300 | | |
| Detector | Gatan K3 | | |
| Magnification (nominal) | 22500 | | |
| Pixel size (Å/pix) | 1.1 | | |
| Flux (e⁻/pix/sec) | 8 | | |
| Frames per exposure | 80 | | |
| Exposure (e⁻/Å²) | 80 | | |
| Defocus range (µm) | 1.0–2.5 | | |
| Micrographs collected | 3203 | | |
| Sample | SARS-CoV-2 S + N3-1 Fab | | |
| **3D reconstruction statistics** | | | |
| | Overall | 2 RBDs +N3-1 | 1 RBD + N3-1 |
| EMDB ID | EMD-41399 | EMD-41374 | EMD-41382 |
| PDB ID | - | 8TM1 | 8TMA |
| Particles | 266,585 | 266,553 | 266,553 |
| Symmetry | C1 | C1 | C1 |
| Map sharpening B-factor | -130.7 | -119.8 | -135.3 |
| Unmasked resolution at 0.143 FSC (Å) | 3.5 | 4.0 | 4.3 |
| Masked resolution at 0.143 FSC (Å) | 2.8 | 2.8 | 3.2 |
| **Model refinement and validation statistics** | | | |
| Composition | | | |
| Amino acids | | 569 | 410 |
| RMSD bonds (Å) | | 0.002 | 0.004 |
| RMSD angles (°) | | 0.64 | 0.68 |
| Average B-factors | | | |
| Amino acids | | 39.0 | 38.7 |
| Ramachandran | | | |
| Favored (%) | | 96.8 | 95.8 |
| Allowed (%) | | 3.2 | 4.0 |
| Outliers (%) | | 0 | 0.2 |
| Rotamer outliers (%) | | 0 | 0 |
| Clash score | | 3.0 | 3.5 |
| Cᵦ outliers (%) | | 0 | 0 |
| CaBLAM outliers (%) | | 1.5 | 2.0 |
| MolProbity score | | 1.29 | 1.43 |
| EMRinger score | | 4.36 | 3.66 |
| CC (mask) | | 0.83 | 0.82 |
| Model masked resolution at 0.5 FSC (Å) | | 3.0 | 3.3 |

bridges, hydrophobic interactions and buried surface areas between Fab and RBDs[43].

**SEC-MALS analysis**. SEC-MALS measurements were conducted at room temperature using an Agilent 1260 Infinity II HPLC system coupled to a Dawn MALS detector and Optilab dRI detector (Wyatt Technology Corp). Proteins were separated on a Wyatt 030S5 SEC column at a flow rate of 0.5 ml/min with PBS. Concentrations were determined using a dn/dc of 0.185 RI. Data was analyzed using the ASTRA 8.2.0 software package (Wyatt Technology Corp).

**Statistics and reproducibility**. Assays were performed as biological replicates in duplicate or triplicate. The appropriate statistical test, sample size and significance threshold are defined in the relevant methods sections and figure legends. No data were excluded from the analyses.

**Reporting summary**. Further information on research design is available in the Nature Portfolio Reporting Summary linked to this article.

## Data availability

Molecular coordinates for N3-1 Fab complexes with SARS-CoV-2 RBDs have been deposited to the Protein Data Bank (PDB ID: **8TM1 and 8TMA**). One global EM map (**EMD-41399**) and two focused refinement maps (**EMD-41374** and **EMD-41382**) are available at the Electron Microscopy Data Bank. Structural data are presented in Fig. 3, Table 1, and Supplementary Figs. 8 and 15-16. Antibody VH sequencing is available at the NCBI Sequence Read Archive (BioProject ID: **PRJNA1037156**). Raw proteomic data and search results have been deposited in PRIDE and ProteomeXchange (**PXD046829**). The source data behind the graphs in the paper can be found in Supplementary Data.

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

## Acknowledgements

We thank Greg Ippolito, Jason Lavinder, and Will Voss for technical assistance. We would like to thank Dr. Sasha Dickinson from the Sauer Structural Biology Laboratory at the University of Texas at Austin for his assistance with microscope data collection. The Sauer Structural Biology Laboratory is supported by the University of Texas College of Natural Sciences and by award RR160023 from the Cancer Prevention and Research Institute of Texas (CPRIT). Funding for USAMRIID was provided through the CARES Act with programmatic oversight from the Military Infectious Diseases Research Program–project 14066041. Opinions, conclusions, interpretations, and recommendations are those of the authors and are not necessarily endorsed by the U.S. Army. The mention of trade names or commercial products does not constitute endorsement or recommendation for use by the Department of the Army or the Department of Defense. Molecular graphics and analyses were performed with UCSF Chimera, developed by the Resource for Biocomputing, Visualization, and Informatics at the University of California, San Francisco, with support from NIH P41-GM103311. SEC-MALS analysis and interpretation was performed by the Biomolecular Characterization Unit, Protein and Monoclonal Antibody Production Core at Baylor College of Medicine under the direction of Josephine C. Ferreon and funding from NIH S10-OD030276. This research was funded in part by: the Army Research Laboratory's TRANSFORME Essential Research Program (JDG, DRB, RR, THSS); a Cooperative Agreement (W911NF-17-2-0091) between ARL and UT Austin to ADE, EMM, and GG; a grant from DTRA (HDTRA12010011) awarded to JDG and ADE; a National Institutes of Health (NIH)/ National Institute of Allergy and Infectious Diseases (NIAID) grant awarded to JSM (R01 AI127521); financial assistance award 70NANB20H037 to JDG from the US Department of Commerce, National Institutes of Standards and Technology (NIST) via the National Institute for Innovation in Manufacturing Biopharmaceuticals (NIIMBL); a Welch grant (F-1016) awarded to IJF; a Welch grant (F-1515) awarded to EMM; funding support from the Huck Institutes of Life Sciences and the Pennsylvania Agricultural Experiment Station (to VK and SK); an NIH grant (R01 AI158177-01) to SK; and an NIH grant (R35 GM122480) to EMM.

## Author contributions

Conceptualization: J.G., C.H., A.P.H., E.C.G., D.R.B., J.S.M., and J.D.G.; Methodology: J.G., C.H., A.P.H., E.C.G., L.Z., F.B., N.W., K.J., A.H., R.R., T.S.S., S.V.K., V.K., W.W., R.S., D.R.B., J.S.M., and J.D.G; Investigation: J.G., C.H., A.P.H., E.C.G., L.Z., F.B., N.W., K.J., A.H., C.T., S.V.K., V.K., S.A., H.X., X.X., P.S., R.R., W.W., R.S., M.B., S.C.W., J.M.M., D.R.B., J.S.M., and J.D.G.; Data Analysis and Interpretation: J.G., C.H., A.P.H., D.R.B., L.Z., N.W., A.H., P.S., H.X., X.X., J.M.D., E.M.M., J.S.M., and J.D.G.; Data Curation: J.G., C.H., A.P.H., E.C.G., L.Z., N.W., D.R.B., J.S.M., X.X., A.H., J.M.D., J.S.M., and J.D.G.; Original Draft: J.G., C.H., A.P.H., E.C.G., D.R.B., J.M.M., J.S.M., V.K., E.M.M., and J.D.G; Review & Editing: J.G., C.H., A.P.H., E.C.G., D.R.B., A.D.E., E.M.M., J.M.M., G.G., J.S.M., and J.D.G.; Funding: J.M.D., V.K., S.V.K., G.G., A.D.E., J.S.M. and J.D.G.

## Competing interests

ADE, EMM, GG, and DRB declare competing financial interests in the form of provisional and granted patent applications relevant to IgSeq. JG, CH, ECG, APH, DRB, EMM, JSM, ADE, GG, and JDG have filed provisional applications for the discovery of neutralizing antibodies. JG, DRB, ECG, APH, and JDG have filed applications for additional methods relevant to this work. All other authors declare no competing interests.

## Ethics

The acquisition of blood specimens from convalescent individuals was approved by the University of Texas at Austin Institutional Review Board (protocol 2020-03-085; Breadth of serum antibody immune responses prior to, or following, patient recovery in asymptomatic and non-severe COVID-19). Informed consent was obtained from all participants. All ethical regulations relevant to human research participants were followed.
