## [Peer Review File · Communications Biology]

Reviewers' comments:

Reviewer #1 (Remarks to the Author):

The ms describes characterization SARS-CoV-2 neutralizing antibody (N3-1) which recognizes the spike protein. A Cryo-EM structure of N3-1 with the original Wuhan strain spike ectodomain in a 2-up, 1-down receptor binding domain configuration shows that it likely binds to 3 receptor binding domains at once. The Omicron variant escapes this antibody, not by specific mutations of RBD residues, but likely by having a conformation with fewer receptor binding domains up, so that the configuration supporting avid antibody binding is less readily adopted. The main importance of the ms is the structural characterization of the novel interaction with spike and the understanding of immune escape from SARS-CoV-2. An interesting extrapolation is that engineering immunogens with multiple states may be important for a broad immune response. Additional structure-based antibody mutants designed to restore binding and neutralization of Omicron may be in line with the proposed escape mechanism. Overall, the study will be interesting to readers interested in SARS-CoV-2 as well as a wide community of biologists.

Major points to address:

1. IgG binding a single spike protein via two Fabs is an important mechanism for neutralization by N3-1 and is consistent with the data in the manuscript indicating that the antibody but not Fabs alone are neutralizing. However, because the high-resolution structure is obtained with Fabs, the structure does not show that two Fabs can bind at once from a single IgG. Supplementary Figure 8 is suggested as evidence (Line 148) for 2 Fabs of an IgG binding at once. If a cryo-EM structure of the IgG bound to the spike cannot be obtained, this point can be supported by further interpretation of the images, e.g. matching computed projections for views in Supplementary Figure 8 confirming to the reader that two Fabs are bound. Alternatively, the authors should confirm for readers that it is really possible to build an antibody model with two Fabs in the proposed binding mode (e.g. no clashes, strain).

2. Have the authors tried to image N3-1 in complex with Omicron spikes, e.g. antibody mutant N3-1_I which neutralizes Omicron BA.1 to confirm the anticipated mode of binding and comparison with the Wuhan binding mode?

3. There are 3 structures reported in this ms, one consensus refinement and two focused refinements. They should be clearly labelled in the columns of Supplementary Table 5. Supplementary Table 5 should report map-model FSC. There should also be 3 FSC curves in Supplementary Figure 15. There should be 3 map or map/model database depositions to support the data. There is only one deposition indicated in the ms.

4. The antibody heavy chain was identified by IgSeq for from convalescent sera and combined with light chains from a public light-chain library. In addition, a yeast surface display approach was used for identifying antibodies against spike. I think the Discussion section should tell us a bit more about the take home message for future studies.

5. I would remove the phrase 'report the discovery' of N3-1 from the abstract, given that the authors have already published some data about N3-1 in Javanmardi et al. 2022 (ref. 31 of this ms).

Other specific points:

1. Readability of the ms would be improved by the text referring to specific panels of the figures more completely, e.g. for structural data in figure 3.

2. Figure 4a and b seem to be swapped, and description of content not completely clear.

Reviewer #2 (Remarks to the Author):

According to the latest WHO report, COVID-19 is no longer a public health emergency of international concern. However, the continuous mini-waves of SARS-CoV-2 (with new lineage, such as XBB.1.16) infections inform us that the vaccine (or antibody) development is still important. Particularly, the broadly vaccine and/or antibody is what we need at this stage. Here, Goike et al. report a neutralizing antibody N3-1 against original SARS-CoV-2. These authors illustrated the unique binding mode of N3-1 in complex with spike proteins, they found that altered RBD up-down equilibria are crucial to omicron escape from N3-1. In all, this study is carefully carried out and provides new information for vaccine designs. I suggested that this paper should be published in Communications Biology.

However, there are several comments should be addressed before acceptance.

Major comment:

Besides the Cryo-EM structure, the author should provide other proofs (such as SEC-MALS, ITC or analytical ultracentrifugation etc) to confirm the molar ratio between N3-1 and spike in solution.

Minor type mistakes:

Page 4, line 128, EC50, "50" should be subscript.

Page 7, line 212, "BA2.12.1" to "BA.2.12.1",
line 218, Kd, should be in italics.

Reviewer #3 (Remarks to the Author):

The establishment of a method for rapidly selecting potent neutralizing antibodies from sera of infected patients reported by Goike et al. in this paper is very useful against new infectious disease threats that may arise in the future. It is very interesting to note that the existence of the quaternary epitope, which is difficult to identify by conventional immunological methods alone, was elucidated by structural analysis. Since the findings obtained in this study will provide an important structural basis for future vaccine development and antibody drug design, I believe that an accurate description of the structure and its evaluation are important, and I therefore request comments and revisions on the following points.

1) In the complex of RBD-down and -up with N3-1 Fabs, it is mentioned that the binding pattern of the two types of N3-1 Fab and RBD-up is the same, so the global refined structure of RBD-up and N3-1 Fab should also be registered in PDB database.

2) How were the atomic interactions, such as hydrogen bonding, salt bridges, hydrophobic interactions, etc or buried surface area. determined, and the use of programs such as Pisa, etc., should be specified in the Methods section. The definition of atomic interactions (bond distances, angles, and other threshold values) should be clearly stated.

3) For example, the definition of the subscript "a" in Arg100a is not given, and its meaning is unclear. Tyr98", "Phe99" and "Arg100a" in H-CDR3 are registered as "Tyr103", "Phe 104" and "Arg106" in PDB (7SIX). Amino acid numbers should be unified to avoid confusion of readers.

4) Structural analysis shows that Fab can bind to "up, down" and "up" of the RBD at the same time, please explain why Fab cannot bind to "up, down" and "down" at the same time. Also, please explain why the complete antibody cannot bind to "up, down" and "down" at the same time. Was binding to "up, down" and "down" etc. found in the process of single-particle structural analysis as a minor state? If so, the percentage information may correlate with the neutralizing activity of the antibody, which could be an indicator if the same type of antibody is obtained in the future, so please add the information to Supplementary Figure 14.

Minor revision

- 1) Figure 3. a is not a Cryo-EM structure but a Cryo-EM density (map).
- 2) The other RBD-up in the upper panel of Figure 3.a should also be visible.
- 3) The color difference between L-CDR3 and H-CDR3 in the zoom panel in Figure 3. should be the same as the color of the density map
- 4) Figure 3.d. The letters Asn487 in the bottom zoom panel do not indicate which amino acids are shown.
- 5) P.7 Please unify the notation such as N3-1_Q. QN3-1 and N3Q-1 are mixed in the figure.
- 6) Figure 4. a and b in legend are reversed
- 7) References such as cisTEM and ISOLDE should be added.

Response to Reviewers

"SARS-COV-2 Omicron variants conformationally escape a novel quaternary antibody binding mode" - COMMSBIO-23-1324-T

We thank the editors, editorial staff, and reviewers for their helpful suggestions and critical reading of the submitted manuscript. Please find below our point-by-point response to the reviewer's feedback.

Reviewers' Comments:

Reviewer #1 (Remarks to the Author):

The ms describes characterization SARS-CoV-2 neutralizing antibody (N3-1) which recognizes the spike protein. A Cryo-EM structure of N3-1 with the original Wuhan strain spike ectodomain in a 2-up, 1-down receptor binding domain configuration shows that it likely binds to 3 receptor binding domains at once. The Omicron variant escapes this antibody, not by specific mutations of RBD residues, but likely by having a conformation with fewer receptor binding domains up, so that the configuration supporting avid antibody binding is less readily adopted. The main importance of the ms is the structural characterization of the novel interaction with spike and the understanding of immune escape from SARS-CoV-2. An interesting extrapolation is that engineering immunogens with multiple states may be important for a broad immune response. Additional structure-based antibody mutants designed to restore binding and neutralization of Omicron may be in line with the proposed escape mechanism. Overall, the study will be interesting to readers interested in SARS-CoV-2 as well as a wide community of biologists.

Major points to address:

1. IgG binding a single spike protein via two Fabs is an important mechanism for neutralization by N3-1 and is consistent with the data in the manuscript indicating that the antibody but not Fabs alone are neutralizing. However, because the high-resolution structure is obtained with Fabs, the structure does not show that two Fabs can bind at once from a single IgG. Supplementary Figure 8 is suggested as evidence (Line 148) for 2 Fabs of an IgG binding at once. If a cryo-EM structure of the IgG bound to the spike cannot be obtained, this point can be supported by further interpretation of the images, e.g. matching computed projections for views in Supplementary Figure 8 confirming to the reader that two Fabs are bound. Alternatively, the authors should confirm for readers that it is really possible to build an antibody model with two Fabs in the proposed binding mode (e.g. no clashes, strain).

RESPONSE: We thank the reviewer for their favorable summary and critical reading of our manuscript. We have generated an antibody model with two Fabs bound to a spike trimer with two RBDs in the up conformation. As the figure below shows, the proposed binding mode could occur without clashes to RBDs. We have added this to Supplementary Figure 8.

2. Have the authors tried to image N3-1 in complex with Omicron spikes, e.g. antibody mutant N3-1_I which neutralizes Omicron BA.1 to confirm the anticipated mode of binding and comparison with the Wuhan binding mode?

RESPONSE: We appreciate the suggestion but we have not performed that additional cryo-EM experiment, which may be complicated by the lower affinity of the antibodies for BA.1.

3. There are 3 structures reported in this ms, one consensus refinement and two focused refinements. They should be clearly labelled in the columns of Supplementary Table 5. Supplementary Table 5 should report map-model FSC. There should also be 3 FSC curves in Supplementary Figure 15. There should be 3 map or map/model database depositions to support the data. There is only one deposition indicated in the ms.

RESPONSE: Thank you for the suggestions. We have revised the labels to the columns in Supplementary Table 5 and updated the FSC curves in the figure (now Supplementary Figure 16). We have also deposited a global map (EMD-41399), two focused maps (EMD-41374; EMD-41382) and the corresponding atomic models (PDB ID: 8TM1; PDBID: 8TMA).

4. The antibody heavy chain was identified by IgSeq for from convalescent sera and combined with light chains from a public light-chain library. In addition, a yeast surface display approach was used for identifying antibodies against spike. I think the Discussion section should tell us a bit more about the take home message for future studies.

RESPONSE: We have revised the first paragraph of the discussion for clarity and included more about the take home message for future studies: “By utilizing both IgSeq and YSD, we queried the serological and cellular repertoires, respectively. Clonal diversity among the antigen-specific peripheral B-cells is distinct from that of the circulating serum antibodies, and mining from both repertoires provides better understanding of an individual's immune response and gives more opportunity to identify strong neutralizing mAbs³¹.”

5. I would remove the phrase ‘report the discovery’ of N3-1 from the abstract, given that the authors have already published some data about N3-1 in Javanmardi et al. 2022 (ref. 31 of this ms).

RESPONSE: We have removed the phrase in the Abstract.

Other specific points:

1. Readability of the ms would be improved by the text referring to specific panels of the figures more completely, e.g. for structural data in figure 3.

RESPONSE: Thank you for the suggestions. We have called out more figure 3 panels to the corresponding structural descriptions in the text.

2. Figure 4a and b seem to be swapped, and description of content not completely clear.

RESPONSE: We have swapped legend labels 4a and b, and we have revised the legend to better describe the figure.

Reviewer #2 (Remarks to the Author):

According to the latest WHO report, COVID-19 is no longer a public health emergency

of international concern. However, the continuous mini-waves of SARS-CoV-2 (with new lineage, such as XBB.1.16) infections inform us that the vaccine (or antibody) development is still important. Particularly, the broadly vaccine and/or antibody is what we need at this stage. Here, Goike et al. report a neutralizing antibody N3-1 against original SARS-CoV-2. These authors illustrated the unique binding mode of N3-1 in complex with spike proteins, they found that altered RBD up-down equilibria are crucial to omicron escape from N3-1. In all, this study is carefully carried out and provides new information for vaccine designs. I suggested that this paper should be published in *Communications Biology*.

However, there are several comments should be addressed before acceptance.

Major comment:

Besides the Cryo-EM structure, the author should provide other proofs (such as SEC-MALS, ITC or analytical ultracentrifugation etc) to confirm the molar ratio between N3-1 and spike in solution.

RESPONSE: We thank the reviewer for their favorable recommendation. We performed SEC-MALS for the N3-1 IgG in complex with SARS-CoV-2 S, and the measured molecular weights support our proposed model (Supplementary Fig. 9). After introducing a 3:1 molar mixture to SEC-MALS, we observed a complex of 1:1 IgG to spike and excess free IgG.

Minor type mistakes:

Page 4, line 128, EC50, "50" should be subscript.

RESPONSE: Thank you for careful review of the manuscript. We have revised line 128 as recommended.

Page 7, line 212, "BA2.12.1" to "BA.2.12.1",
line 218, Kd, should be in italics.

RESPONSE: We have revised lines 212 and 218 as recommended.

Reviewer #3 (Remarks to the Author):

The establishment of a method for rapidly selecting potent neutralizing antibodies from sera of infected patients reported by Goike et al. in this paper is very useful against new infectious disease threats that may arise in the future. It is very interesting to note that the existence of the quaternary epitope, which is difficult to identify by conventional immunological methods alone, was elucidated by structural analysis. Since the findings obtained in this study will provide an important structural basis for future vaccine development and antibody drug design, I believe that an accurate description of the

structure and its evaluation are important, and I therefore request comments and revisions on the following points.

1) In the complex of RBD-down and -up with N3-1 Fabs, it is mentioned that the binding pattern of the two types of N3-1 Fab and RBD-up is the same, so the global refined structure of RBD-up and N3-1 Fab should also be registered in PDB database.

RESPONSE: We thank the reviewer for their suggestions. We have deposited both focused maps (EMD-41374; EMD-41382) and the corresponding atomic models (PDB ID: 8TM1; PDB ID: 8TMA) in EMDD and PDB database. The cryo-EM maps are now shown in Figure 3b and 3d.

2) How were the atomic interactions, such as hydrogen bonding, salt bridges, hydrophobic interactions, etc or buried surface area. determined, and the use of programs such as Pisa, etc., should be specified in the Methods section. The definition of atomic interactions (bond distances, angles, and other threshold values) should be clearly stated.

RESPONSE: We used PISA for determining atomic interactions and have added these specifics to the Methods section.

3) For example, the definition of the subscript "a" in Arg100a is not given, and its meaning is unclear. Tyr98", "Phe99" and "Arg100a" in H-CDR3 are registered as "Tyr103", "Phe 104" and "Arg106" in PDB (7SIX). Amino acid numbers should be unified to avoid confusion of readers.

RESPONSE: We have replaced the PDB entry with the Kabat-numbered version (PDB IDs 8TM1 and 8TMA) to match the manuscript. Additionally, we now state our use of the Kabat numbering scheme in the main text, including the text "residues 93-99,100a-100j,101-102 in the Kabat antibody numbering scheme" in our definition of the N3 CDRH3.

4) Structural analysis shows that Fab can bind to "up, down" and "up" of the RBD at the same time, please explain why Fab cannot bind to "up, down" and "down" at the same time. Also, please explain why the complete antibody cannot bind to "up, down" and "down" at the same time. Was binding to "up, down" and "down" etc. found in the process of single-particle structural analysis as a minor state? If so, the percentage information may correlate with the neutralizing activity of the antibody, which could be an indicator if the same type of antibody is obtained in the future, so please add the information to Supplementary Figure 14.

RESPONSE: We superimposed the N3-1 Fab bound to RBD structure from this study with apo spike structure that has two RBDs in the down conformation (PDB: 6XKL). As the figure shows below, in order to let N3-1 Fab bind to RBD-up,

N3-1 heavy chain will inevitably clash if the other two RBDs are in the down conformation. We have added this to Supplementary Figure 8 and discussed briefly in the main text. We did not observe, even as a minor state, N3-1 binding to "up, down" and "down" in the single-particle structural analysis.

Minor revision

1) Figure 3. a is not a Cryo-EM structure but a Cryo-EM density (map).

RESPONSE: We have corrected the figure legend as recommended.

2) The other RBD-up in the upper panel of Figure 3.a should also be visible.

RESPONSE: We have added a lower resolution contour to help visualize the interaction of both Fabs with RBDs.

3) *The color difference between L-CDR3 and H-CDR3 in the zoom panel in Figure 3. should be the same as the color of the density map*

RESPONSE: We have matched the colors in the cryo-EM maps and the atomic models.

4) *Figure 3.d. The letters Asn487 in the bottom zoom panel do not indicate which amino acids are shown.*

RESPONSE: We have indicated Asn487 in the figure (now Fig 3e).

5) *P.7 Please unify the notation such as N3-1_Q. QN3-1 and N3Q-1 are mixed in the figure.*

RESPONSE: We have revised the N3-1 mutation names in Fig. 3 and Supplementary Figs. 12-13 for consistency with the main text.

6) *Figure 4. a and b in legend are reversed*

RESPONSE: We have corrected this error.

7) *References such as cisTEM and ISOLDE should be added.*

RESPONSE: We have added the references.

REVIEWERS' COMMENTS:

Reviewer #1 (Remarks to the Author):

The revision has successfully addressed almost all points raised. As previously requested, Supplementary Table 5 should include a resolution assessment based on the map-model FSC (the resolution at which the map-model FSC crosses the 0.5 threshold) for the two structures for which models were built.

Reviewer #2 (Remarks to the Author):

The authors answered all my comments, and they confirmed the molar ration between IgG and Spike by the SEC-MALS assay. I support the publication of this revised manuscript.

Reviewer #3 (Remarks to the Author):

I've reviewed the revised manuscript, and I'm pleased to see that the author has made substantial improvements based on the feedback. The revisions have significantly enhanced the manuscript's quality and clarity.

Given the thorough revisions, I recommend accepting the manuscript for publication in Communications biology. I believe it will contribute positively to the field.

I appreciate the opportunity to review this manuscript and believe it is now ready for publication.